# Arctic-Embed 2.0: Multilingual Retrieval Without Compromise

**Puxuan Yu**
Snowflake Inc.
puxuan.yu@snowflake.com

**Luke Merrick**[*]
DatologyAI
contact@lukemerrick.com

**Gaurav Nuti**
Snowflake Inc.
gaurav.nuti@snowflake.com

**Daniel Campos**
Snowflake Inc.
daniel.campos@snowflake.com

## Abstract

This paper presents the training methodology of Snowflake Arctic-Embed 2.0, a set of open-source text embedding models built for effective and efficient multilingual retrieval. While prior works have suffered from degraded English retrieval quality, Arctic-Embed 2.0 delivers competitive retrieval quality on multilingual and English-only benchmarks, and supports Matryoshka Representation Learning (MRL) for efficient embedding storage with significantly lower compressed quality degradation compared to alternatives. Beyond describing the design and implementation details, we highlight critical research questions encountered during development, including the mechanisms of cross-lingual transfer in retrieval pre-training and what we term the "English performance gap" - the systematic quality difference between specialized English-only models and multilingual alternatives. Through targeted experiments addressing these questions, we derive insights from both positive and negative results, contributing to a broader understanding of multilingual embedding models and aiming to stimulate further research on improving cross-lingual representation quality while maintaining strong monolingual performance.

## 1 Introduction

Transformer-based embedding models have become a cornerstone of various information retrieval (IR) applications (e.g., search engines and retrieval-augmented generation systems). Although many efforts have focused on English-only retrieval (Merrick et al., 2024; Wang et al., 2024a; Günther et al., 2024; Nussbaum et al., 2024), considerable efforts have also been directed toward developing multilingual embedding models (Wang et al., 2024b; Zhang et al., 2024; Chen et al., 2024; Sturua et al., 2024). By learning to map queries and documents from multiple languages into a shared representation space, these multilingual text embedding models enable non-English monolingual retrieval as well as cross-lingual retrieval.

We develop Arctic-Embed 2.0[1] to deliver frontier retrieval quality while addressing two key limitations observed in current multilingual embedding models:

**Efficiency Losses**: Many state-of-the-art models that deliver high retrieval effectiveness require a large number of parameters and generate large embedding vectors (Wang et al., 2024a; Sturua et al., 2024). These increase both computational and economic costs of dense retrieval, presenting challenges when dealing with large corpora.

---

[*]Work done while at Snowflake.

[1]The open-source model weights are available under the Apache 2.0 License: snowflake-arctic-embed-m-v2.0 and snowflake-arctic-embed-l-v2.0.

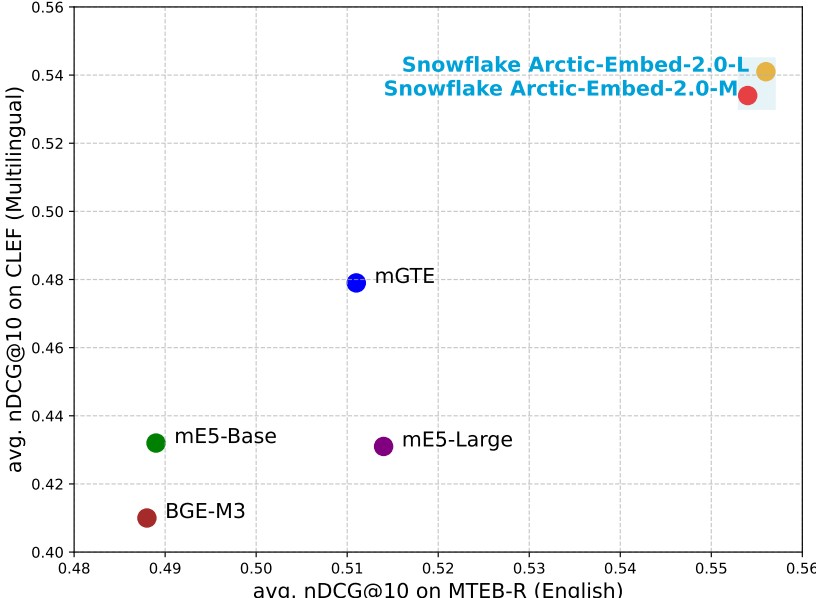

Figure 1: Single-vector dense retrieval performance of open-source multilingual embedding models with fewer than 1B parameters. Scores are average nDCG@10 on MTEB Retrieval (Muennighoff et al., 2023) and the subset of CLEF (ELRA, 2006) covering English, French, Spanish, Italian, and German.

**Compromised English Retrieval Quality**: It is common for multilingual models to underperform their English-only counterparts on English retrieval evaluations (e.g., MTEB Retrieval (Muennighoff et al., 2023)). This trade-off has been a significant pain point in deploying multilingual systems.

Arctic-Embed 2.0 outperforms leading open-source alternatives, making it a highly versatile solution for both English and non-English contexts. Additionally, our two-stage approach to integrating Matryoshka Representation Learning (MRL) (Kusupati et al., 2022) drastically mitigates quality degradation during compression compared to other models supporting dimensionality reduction. Our contributions are as follows:

- We introduce Arctic-Embed 2.0, multilingual embedding models that achieve competitive retrieval quality on both English and multilingual benchmarks and support size-efficient embeddings via MRL.

- We investigate potential causes of reduced English retrieval quality in prior multilingual models. We empirically refute the hypothesis that pretraining on distant languages harms English performance, and we propose alternative explanations for future investigation.

- We reveal that pretrained checkpoint evaluations often fail to predict fine-tuned performance, highlighting the need for improved pretraining evaluation methods.

- We show that while fine-tuning generally enhances cross-lingual transfer, contrastive pretraining in multilingual settings can lead to negative cross-lingual transfer.

## 2 Methodology

We follow a three-stage training framework inspired by prior works (Merrick et al., 2024; Wang et al., 2024b; Nussbaum et al., 2024; Chen et al., 2024; Zhang et al., 2024): pretraining via masked language modeling, contrastive pretraining, and contrastive finetuning.

## 2.1 Masked Language Modeling

We use two open-source pretrained encoder models: `gte-multilingual-mlm-base` (Zhang et al., 2024) for medium size and `bge-m3-retromae` (Chen et al., 2024) for large size. Both models employ the XLM-R tokenizer (Conneau et al., 2020). Further details on the selection of these base models can be found in Appendix A.

## 2.2 Contrastive Training Data

Guided by end-user applications, we focused on European languages: English, French, Spanish, German, Italian, Portuguese, and Polish.

Contrastive pretraining data consists of loosely related pairs that are not human-labeled, such as title-webpage pairs. For English data, we followed Arctic-Embed (Merrick et al., 2024). For multilingual data, we used mC4 (Habernal et al., 2016), CC News (treating page titles as queries and bodies as documents), and multilingual Wikipedia (titles and section headings concatenated as queries, section texts as documents) following Wang et al. (2024b). NLLB (Team et al., 2022) was excluded due to its limited resemblance to query-document tasks and negligible empirical benefit in early tests. For mC4, CC News, and mWiki, we subsetted to our target languages, including English.

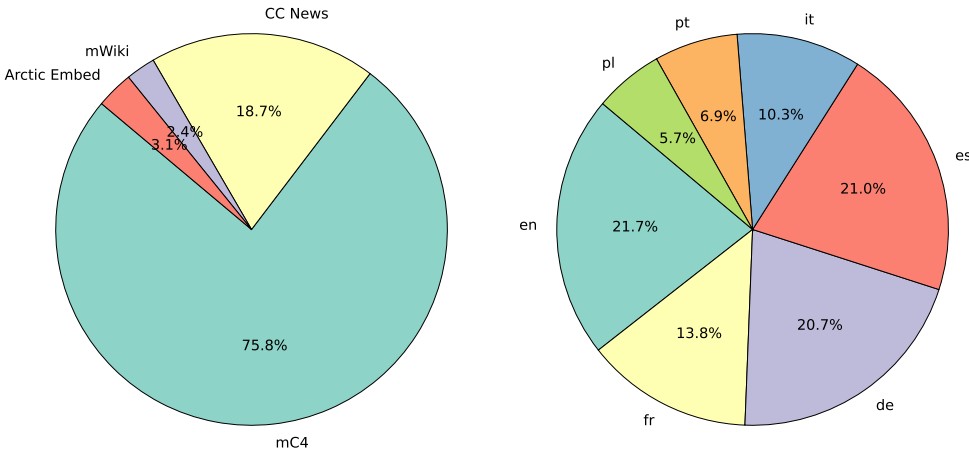

Figure 2: Breakdown of 1.41B contrastive pretraining samples by data source and language.

For finetuning data, we follow the data mix of Merrick et al. (2024) for English, adding MIRACL (Zhang et al., 2023b) training set for high-quality multilingual training samples. We exclude Mr. Tydi (Zhang et al., 2021) due to overlap with MIRACL but use all MIRACL languages (not just target ones), as we observe no negative transfer to retrieval in target languages.

## 2.3 Methods of Data Filtering and Training

We apply heuristic and consistency quality filters to English-only pretraining data. For multilingual pretraining data, we adopt retrieval-based consistency filtering as in several prior works (Nussbaum et al., 2024; Günther et al., 2023; Wang et al., 2024a; Dai et al., 2022), using the small multilingual E5 model (Wang et al., 2024b) to embed queries and documents. Each dataset is partitioned into even shards of approximately 3 million query-document pairs, and pairs are filtered out if the pair's document ranks below rank 20 by vector similarity within all documents in its shard. This results in approximately 1.41 billion *unsupervised* query-document pairs. A detailed breakdown of this combined dataset by source and language is shown in Figure 2.

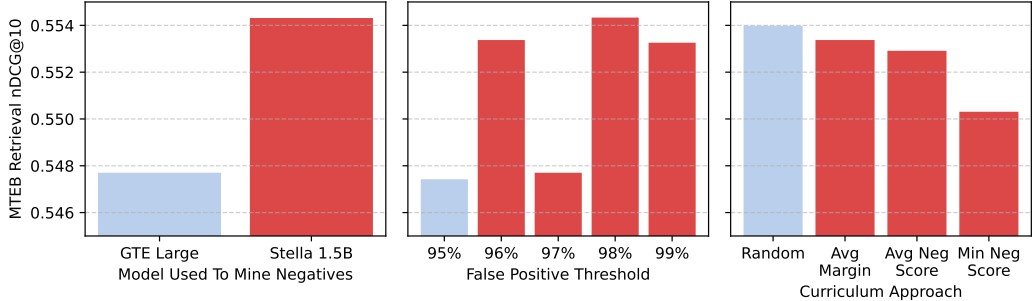

Figure 3: Hard-negative mining ablation studies. A stronger teacher embedding model and well-tuned false-positive cutoff led to improved downstream performance, while a random order of examples performed just as well as various approaches to creating easy-to-hard curricula.

For pretraining, we use the standard InfoNCE contrastive loss (van den Oord et al., 2018) with a temperature $\tau = 0.02$ as our contrastive pretraining objective. We use random in-batch negatives and follow the approach of Nussbaum et al. (2024) and Merrick et al. (2024), sampling all training mini-batches from a single data source at a time. For multilingual sources, different-language subsets are treated as distinct sources during batch construction. We use a batch size of 32,768, a maximum query length of 32, and a maximum document length of 256. We use peak learning rates of 3e-5 and 1e-4 for the large and medium models, respectively, following a linear warmup-stable-decay (WSD) schedule (Hu et al., 2024) for 3 epochs. To accommodate the large batch size and dataset scale, we employ activation checkpointing and use 32 H100 GPUs in a distributed data-parallel training setup.

In the finetuning stage, we train using the same InfoNCE loss function but rely on smaller, high-quality datasets and carefully curated negatives instead of random in-batch negatives. We also extend the maximum sequence length for queries and documents to 512 tokens, adjusting the batch size to 256 sets of 1 query, 1 positive doc, and 10 negative docs, changing the learning rate to 1e-5 and 5e-6 for medium and large models, respectively, and adjusting our WSD learning rate schedule to have no warmup and perform linear decay for 6,000 out of a total of 9,342 steps.

## 2.4 Hard Negative Mining for Finetuning

To select the most effective hard negatives, we adopt the strategy from NV Retriever (Moreira et al., 2024): documents scored as most relevant by a teacher embedding model are used as negatives, but any negative with a relevance score exceeding a specified percentage of the known-positive's score is discarded as a potential false negative. We used `stella-en-1.5B-v5`[2] as the English teacher model, and `multilingual-e5-large` for multilingual data. Using `gte-large-en-v1.5` for comparison, we confirm Moreira et al. (2024)'s finding that stronger teacher models yield higher-quality fine-tuning datasets (Figure 3, left). Rather than adhering to the 95% false-positive filtering threshold suggested in prior work, however, we experimented with varying thresholds and observed improvement at higher thresholds (Figure 3, middle).

We also explored curriculum learning for hard negative mining, inspired by Merrick et al. (2024). Specifically, we ordered data by increasing negative hardness during training, using metrics like the average margin between relevance scores of negatives and known-positives, average negative relevance score, and minimum negative relevance score. However, as shown in Figure 3 (right), random data ordering produced comparable or better results than any curriculum-based approach.

---

[2] https://huggingface.co/dunzhang/stella_en_1.5B_v5

| Model Name | Multilingual? | #Params | Emb. Dim. | MTEB-R | CLEF |
|---|---|---|---|---|---|
| E5 Base v2 | no | 86M | 768 | 0.502 | - |
| ME5 Base | yes | 86M | 768 | 0.489 | 0.432 |
| GTE Base En v1.5 | no | 113M | 768 | 0.540 | - |
| GTE Multilingual Base | yes | 113M | 768 | 0.511 | 0.479 |
| Arctic-Embed 1.0-M | no | 86M | 768 | 0.549 | - |
| Arctic-Embed 2.0-M | yes | 113M | 768 | **0.554** | **0.534** |
|   + truncation | yes | 113M | 256 | 0.549 | 0.522 |
| E5 Large v2 | no | 303M | 1024 | 0.506 | - |
| ME5 Large | yes | 303M | 1024 | 0.514 | 0.431 |
| BGE Large En | no | 303M | 1024 | 0.521 | - |
| BGE M3 | yes | 303M | 1024 | 0.488 | 0.410 |
| Arctic-Embed 1.0-L | no | 303M | 1024 | **0.560** | - |
| Arctic-Embed 2.0-L | yes | 303M | 1024 | 0.556 | **0.541** |
|   + truncation | yes | 303M | 256 | 0.547 | 0.530 |
| OpenAI Text Emb. 3 Large | yes | unknown | 3072 | 0.554* | 0.565 |
|   + truncation | yes | unknown | 256 | 0.517* | 0.510 |
| Google Text Emb. 4 | no | 1.2B | 768 | 0.557* | - |
|   + truncation | no | 1.2B | 256 | 0.524* | - |
| Voyage Multilingual 2 | yes | unknown | 1024 | - | 0.569 |

Table 1: nDCG@10 performance of models in the evaluations, grouped by size. The best-performing model in each group is highlighted in **bold**, while the second-best is underlined. #Params: the count of non-embedding parameters, except for Google's model, which only reports total parameters. Asterisks denote results from Lee et al. (2024).

## 2.5 Matryoshka Representation Learning

While the modest model sizes of our models enable inference with low latency and high throughput on modern GPU hardware, scalability in downstream retrieval systems often depends on optimizing the memory footprint of embedding vectors, since retrieval costs typically scale with the total memory consumed by embeddings (Aguerrebere et al., 2023). Merrick (2024) showed that combining MRL (Kusupati et al., 2022) with scalar quantization is an effective method for compressing embeddings with minimal retrieval degradation. We emulate this approach, applying MRL loss during both pretraining and finetuning stages at a single truncated dimensionality of 256. This enables the medium and large models to achieve 3x and 4x compression, respectively, while ensuring a homogeneous distribution of components that facilitates aggressive quantization for further compression.

## 3 Benchmarking

### 3.1 Evaluation Data Sets

We evaluate English-only and multilingual retrieval using the widely adopted MTEB Retrieval benchmark (Muennighoff et al., 2023), and several languages from the CLEF 2000-2003 test suite (ELRA, 2006). Details of CLEF can be found in Appendix B. We choose CLEF over MIRACL (Zhang et al., 2023b) because MIRACL relies exclusively on multilingual Wikipedia, and its training dataset is widely used for both language model (LM) pretraining and embedding model training (including ours). Consequently, MIRACL does not effectively assess a model's ability to generalize beyond the Wikipedia domain. In contrast, the CLEF dataset, lacking a training set and not being freely accessible online, is derived from the news domain, making it a crucial tool for evaluating out-of-domain multilingual retrieval. We still present and discuss results on MIRACL in Appendix C. For per-task performance on MTEB Retrieval, see Appendix D.

## 3.2 Results

The evaluation results, measured in nDCG@10, for MTEB-R and CLEF are presented in Table 1. We run all open-source models ourselves while including some results published by Lee et al. (2024) for closed-source models. We additionally run OpenAI's `text-embedding-3-large` model and Voyage AI's `voyage-multilingual-2` models on CLEF for additional points of reference.

Overall, our models achieve the best performance in their respective size categories on MTEB-R and CLEF, consistently outperforming same-sized competitors and delivering results comparable to flagship closed-source offerings while being at least a magnitude smaller.

Among models trained with MRL, Arctic-Embed 2.0 pulls ahead as the clear leader when embeddings are truncated to 256 dimensions. In this setting, our medium size model outscores the best competitor (Google `text-embedding-004`) 0.549 to 0.524 on MTEB-R despite having far fewer model parameters, and our medium and large models retain 99% and 98% of the original MTEB-R scores, respectively – substantially better than Google `text-embedding-004` (94%) (Lee et al., 2024) and OpenAI Text Embedding 3 Large (93%) (OpenAI, 2024). We postulate that this stronger relative performance under truncation is a result of applying MRL to contrastive *pretraining* as well as contrastive finetuning, as Lee et al. (2024) indicate that MRL was only applied in the finetuning stage for Google's `text-embedding-004` model.

# 4 Research Questions From The Journey

Several empirical observations arose during the development of Arctic-Embed 2.0 which lead to interesting and relevant research questions. Here we present two of these research questions which we explored through additional experimentation. Though our experiments shed some light on the situation, both questions remain open as important lines for future study.

## 4.1 RQ1: Cross-lingual Transfer

**How much does our large-scale contrastive pretraining benefit retrieval for languages not represented in the pretraining data?** Cross-lingual transfer (CLT) is a phenomenon wherein language-agnostic task knowledge is transferred from resource-rich source languages to target languages with limited or no resources. After observing strong scores across the full MIRACL benchmark (including on languages not covered by our contrastive pretraining), we focus on CLT in pretraining, though this phenomenon has also been studied in multilingual retrieval during the finetuning stage (Zhang et al., 2023a).

### 4.1.1 Experiments

We evaluate checkpoints of our medium model during contrastive pretraining at 2K-step intervals up to 10K steps, then at 10K-step intervals thereafter, assessing their performance on MIRACL. We evaluate in two ways: (1) direct evaluation of the checkpoint, and (2) finetune the checkpoint, then evaluate it.

### 4.1.2 Results

The evaluation results on MIRACL are shown in Figure 4. Results with and without finetuning are represented by solid and dotted lines, respectively. From this plot, we observe the following:

**Evaluation before finetuning can be misleading.** For instance, without finetuning, the 130K-step checkpoint appears 18.3% worse than the 8K-step checkpoint on MIRACL's English subset. After finetuning, however, it performs 3.1% better.

**We find little CLT in contrastive pretraining.** We observe negative trends in pre- and post-finetuned evaluation scores across most language families beyond those represented

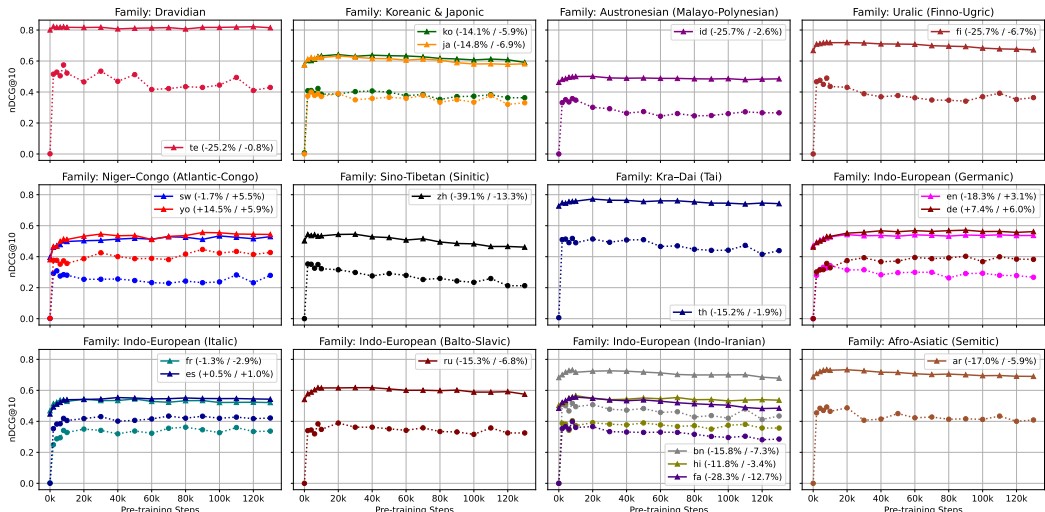

Figure 4: MIRACL performance (in nDCG@10) at different points during contrastive pre-training. Languages are grouped by linguistic families provided by Zhang et al. (2023b). Dotted lines represent non-finetuned runs, while solid lines represent finetuned runs. The relative improvement or deterioration of model performance at the end (130K steps) compared to the 8K-step checkpoint is reported for both non-finetuned and finetuned runs.

by the pretraining data. On evaluations with finetuning, the benefits of pretraining appear to peak within the first 10K steps, after which performance begins to decline for languages not represented by the pretraining data. We observe negative CLT effects particularly in Chinese (-13.3%), Japanese (-6.9%), Russian (-6.8%), Finnish (-6.7%), Korean (-5.9%), and, surprisingly, French (-2.9%), which makes up 13.8% of the pretraining data (Figure 2).

## 4.2 RQ2: English Performance Gap

**Why do many multilingual embedding models perform worse on English retrieval than English-only variants?** As shown in Table 1, transitioning from English-only to multilingual models results in drops of 1.3, 2.9, and 3.3 points on MTEB-R for E5 Base, GTE Base, and BGE Large, respectively, with several closed-source embedding models providers also providing paired models with similar score gaps.[3] However, despite this precedent, we observe strong English-language retrieval quality in our models. To understand why the degradation seen in other works appears absent in our results, we first conduct pilot experiments (Appendix E) to confirm that we do not observe this language gap in our training.

**Initial hypothesis.** Unable to induce an English score gap in our training regimen (see Appendix E), we look to the multilingual pretraining data used by other works to explain their English score gaps. Since our training data focus on European languages and we observe a negative transfer to certain non-European languages like Chinese in our RQ1 experiments, we hypothesize that certain languages may act as "adversaries" to English in retrieval tasks (i.e., training on these languages strongly diminishes English-language retrieval performance and vice versa).

**Experimental design.** To test this hypothesis, we paired English pretraining data with data from German, Spanish (controls), or Chinese (treatment). For each language, we sampled 600 batches (19.6M examples) from their respective corpora: web crawl for English, CC News for Spanish and German, and C-MTP (Xiao et al., 2024) for Chinese. An English-only run was also trained for 16 epochs, while paired runs were trained for 8 epochs, totaling 314M samples per run. The English-only run was evaluated at the midpoint ("en") to isolate

---

[3]At the time of writing, Google, Voyage AI, and Cohere offer such English-multilingual model pairs.

*data addition* effects, and at completion ("en+en") to examine *partial data replacement*. All runs were fine-tuned on the same data before evaluation.

**Results and analysis.** Figure 5 presents the outcomes of these experiments. Starting with the MTEB-R benchmark, it is evident that incorporating the Chinese pretraining data actually *improves* English retrieval performance, whether as an addition to or a partial replacement of English data. This finding directly contradicts our initial hypothesis. Additionally, we see that this Chinese pretraining data outperforms Spanish and German CC News in improving retrieval across MTEB-R, MIRACL, and, in some cases, CLEF, which notably includes evaluation datasets for German and Spanish but not Chinese. Finally, we note that these findings corroborate the trend observed in Figure 4, where repeated training on English pretraining data primarily benefits English retrieval but provides limited improvement for other languages.

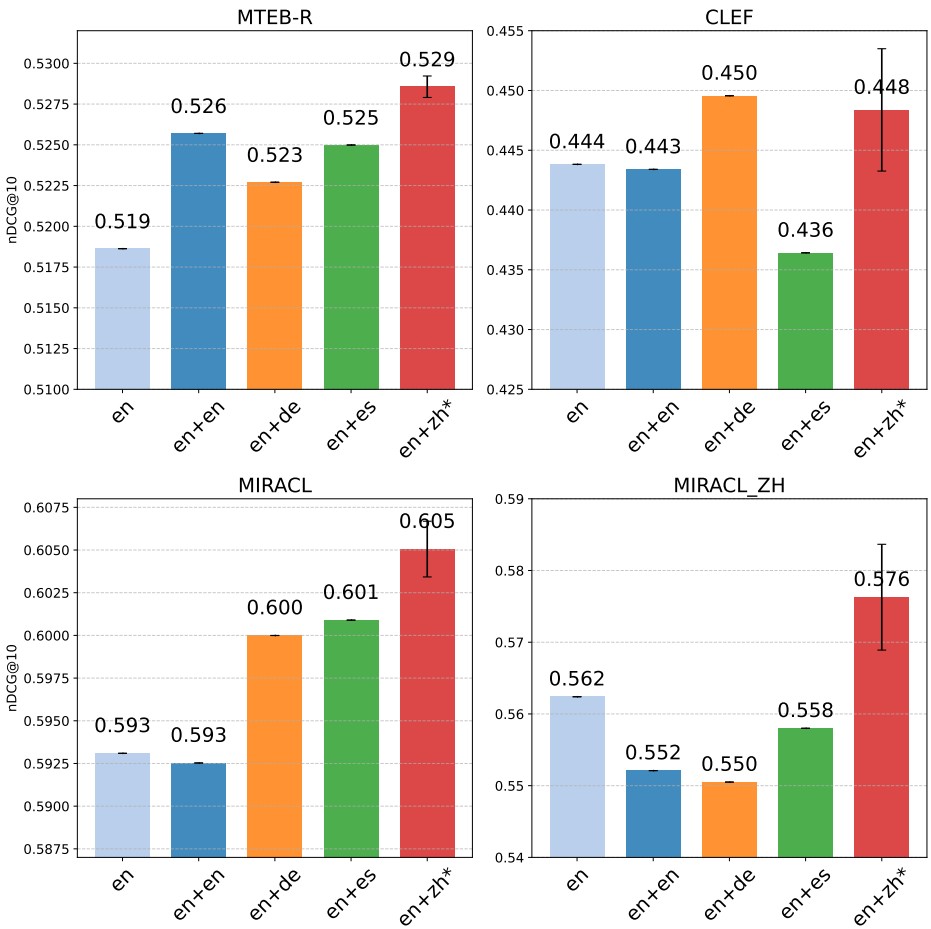

Figure 5: The impact of adding equal amounts of English, German, Spanish, or Chinese data to the existing English pretraining baseline ("en") on downstream retrieval performance. For Chinese data, error bars indicate the standard deviation across consistency filtering levels (top-{1, 5, 10, 20, 30} out of 3M), reflecting the effect of *varying data quality*.

**Alternative hypotheses.** One hypothesis suggested by our results is that data quality plays a more critical role than language itself, with lower-quality multilingual training data accounting for much of the performance gap observed in prior works. Another hypothesis relates to model capacity: the issue may not stem from specific languages but rather from the total number of languages trained simultaneously, potentially exceeding the model's capacity. This could lead to a trade-off where English performance is marginalized to achieve slight improvements across many other languages.

### 4.3 Reflections On The Journey

To summarize some key findings from our model development process, probing experiments, and our takes on interesting and important future directions:

**Data quality matters more than quantity.** We follow the advice of Merrick et al. (2024) to emphasize data quality, deliberately rejecting lower-quality multilingual training data sources, performing retrieval-based consistency filtering, and carefully mining the best negative examples possible for fine-tuning. Though we do not extensively study lower-quality data, we rule out several other possible causes of lower retrieval performance observed in other works, and so we hypothesize that it is this focus on quality that explains why we do not observe degradation in English retrieval performance. In other words, **the English score gap observed in other multilingual models may simply reflect the challenge of acquiring high-quality retrieval training data in certain non-English languages**. Numerous empirical results from this paper lend credence to this quality-centric view, including our finetuning ablations in Section 2.4 and the strong results across languages under a reduced training budget in Appendix E.

**No clear formula for successful cross-lingual transfer in multilingual retrieval models.** In this work, we take a step toward improving the generalization of multilingual embedding models to unseen languages and domains, particularly by evaluating non-English retrieval beyond Wikipedia using CLEF (ELRA, 2006). While we demonstrate the potential for multilingual training to enhance existing benchmark scores (e.g., better MTEB Retrieval scores with multilingual approaches in Appendix E) and expand language support without penalty (e.g., overall benchmark improvements from adding Chinese in Figure 5), the actual process of model development is constrained by several factors: data quality (which is challenging to quantify), the availability of out-of-domain retrieval evaluation benchmarks in more languages, and the need to avoid exceeding model capacity (a concept still not fully understood). Faced with these limitations and uncertainties, incrementally and carefully incorporating non-English data into a proven English-language training mix has turned out to be the most effective strategy available for training useful multilingual embedding models.

**Model "knowledge" and task-calibration are both important yet possibly orthogonal.** As evidenced by the declining un-finetuned English retrieval scores in Figure 4, it is entirely possible for pretraining to show the embedding model hundreds of millions more highly-filtered query-document pairs yet actually induce a *degradation* in downstream retrieval after a certain point (20K steps out of 130K). Add on the finetuning step, however, and we find the performance trend reversed, with an extended pretraining regimen responsible for driving a 3% *increase* to the final nDCG@10 score! This flip-flop not only highlights the importance of measuring final downstream performance, but also demonstrates the intriguing possibility of some amount of useful knowledge being imparted "silently" into the model by contrastive training (somewhat analogously to how non-contrastive MLM pretraining improves the language model without making it fit for retrieval out-of-the-box). In hindsight, it appears that the success of Arctic-Embed 2.0 may stem from both giving the model a substantial mount of "knowledge" through large-scale MLM and contrastive pretraining steps and from carefully "recalibrating" the model with the best and most denoised positive and hard-negative examples.

## 5 Conclusion

In addition to detailing the training process for Arctic-Embed 2.0, we investigate linguistic transfer in embedding models, revealing that prolonged contrastive pretraining does not always enhance cross-lingual transfer, though high-quality pretraining data from languages distant to English can surprisingly do so in some settings. We further discuss how these experiments also uncover concrete evidence of the finetuning step of training "reversing" the negative impact of prolonged pretraining on downstream retrieval performance, a surprising result that indicates new direction for future scientific inquiry.

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

| Language | # Q | # D | # Rels | # Rel/Q |
|---|---|---|---|---|
| English | 246 | 113,005 | 4,769 | 19.4 |
| French | 185 | 129,689 | 3,022 | 16.3 |
| Italian | 176 | 144,040 | 2,626 | 14.9 |
| German | 184 | 153,496 | 3,066 | 16.7 |
| Spanish | 156 | 452,027 | 5,759 | 36.9 |

Table 2: Statistics of the CLEF evaluation datasets: number of queries (# Q), corpus size (# D), number of relevance judgments (# Rels), and average annotations per query (# Rel/Q).

| Model Name | #Params | Emb. Dim. | MIRACL | MIRACL-O | CLEF |
|---|---|---|---|---|---|
| ME5 Base | 86M | 768 | 0.608 | 0.509 | 0.432 |
| GTE Multilingual Base | 113M | 768 | **0.621** | 0.523 | 0.479 |
| Arctic-Embed 2.0-M | 113M | 768 | 0.592 | **0.552** | **0.534** |
|   + truncation | 113M | 256 | 0.578 | 0.545 | 0.522 |
| ME5 Large | 303M | 1024 | 0.651 | 0.540 | 0.431 |
| BGE M3 | 303M | 1024 | **0.678** | **0.568** | 0.410 |
| Arctic-Embed 2.0-L | 303M | 1024 | 0.649 | 0.558 | **0.541** |
|   + truncation | 303M | 256 | 0.638 | 0.547 | 0.530 |
| OpenAI Text Emb. 3 Large | unknown | 3072 | 0.549* | - | 0.565 |
| Google Text Emb. 4 Multilingual | 1.2B | 768 | 0.562* | - | - |
| Voyage Multilingual 2 | unknown | 1024 | - | - | 0.569 |

Table 3: nDCG@10 performance of models on MIRACL grouped by size. Asterisks denote results from Lee et al. (2024).

# A    Selection of Base Models

We aim to develop two model sizes to balance efficiency and effectiveness trade-offs. Based on pilot experiments with small-scale data, we selected the MLM-pretrained m-GTE checkpoint (Zhang et al., 2024) to initialize our medium model and the RetroMAE-pretrained BGE-M3 checkpoint (Chen et al., 2024) for the large model. The m-GTE architecture leverages unpadding and xFormers acceleration (Lefaudeux et al., 2022) for greater efficiency, while the BGE-M3 model benefits from RetroMAE pretraining (Xiao et al., 2022), a retrieval-focused objective critical to its performance.

Both checkpoints use the XLM-R tokenizer. We also tested the newer Llama3 tokenizer (Dubey et al., 2024), designed for multilingual large language models. To accommodate this, we randomly reinitialized model weights, including the embedding layer, and pretrained both models directly without MLM pretraining. However, the Llama3 tokenizer showed no significant effectiveness gains and added efficiency overhead.

# B    CLEF Dataset Details

The statistics of the CLEF datasets used to evaluate the out-of-domain generalizability of multilingual models are reported in Table 2. These datasets have been widely adopted in the literature on non-English monolingual retrieval (Huang et al., 2023; 2024) and cross-lingual retrieval (Yu & Allan, 2020; Yu et al., 2021; Nair et al., 2023) as a reliable benchmark. Since CLEF includes long documents beyond 512 tokens, we enable the maximum token limit for all models during evaluation on this dataset – 512 tokens for E5, 8192 tokens for all other models.

## C    Evaluation Results on MIRACL

Built upon the multilingual Wikipedia collection, a corpus widely used for language model pretraining, the MIRACL multilingual retrieval benchmark (Zhang et al., 2023b) presents a scenario susceptible to in-domain testing and even data leakage, particularly due to its substantial training set.

In Table 3, we compare multilingual models' performance on the full MIRACL set (18 languages), MIRACL-O (an English, French, Spanish, and German subset overlapping with languages in our training data), and the average CLEF performance as a reference point. Our results indicate that while Arctic-Embed 2.0 lags behind the top open-source models by roughly 3 points on the general MIRACL benchmark, it maintains a competitive standing on both the languages included in our training (as reflected in MIRACL-O) and domains outside our training (as reflected in CLEF). Furthermore, the "low on MIRACL but high on CLEF" phenomenon observed in closed-source multilingual models suggests a potential overfitting issue among our open-source competitors specifically towards the MIRACL dataset, which could result in poorer generalization to unseen data.

## D    Per-dataset Results on MTEB Retrieval

| Dataset | AE2L | AE2M | mGTE | BGE-M3 | mE5L | mE5B | mE5S | AE1.5 |
|---|---|---|---|---|---|---|---|---|
| Average | **0.5565** | 0.5538 | 0.5107 | 0.4878 | 0.5140 | 0.4887 | 0.4646 | 0.5509 |
| ArguAna | **0.5917** | 0.5803 | 0.5845 | 0.4411 | 0.5437 | 0.4421 | 0.3910 | 0.5953 |
| ClimateFEVER | 0.4351 | **0.4381** | 0.3514 | 0.2971 | 0.2572 | 0.2326 | 0.2170 | 0.3686 |
| CQADupStack | 0.4610 | **0.4720** | 0.4308 | 0.3995 | 0.3967 | 0.4003 | 0.3706 | 0.4499 |
| DBPedia | 0.4338 | 0.4388 | 0.3972 | 0.3402 | 0.3907 | 0.3700 | 0.3757 | **0.4556** |
| FEVER | 0.9192 | 0.9155 | **0.9206** | 0.8094 | 0.8280 | 0.7942 | 0.7526 | 0.8841 |
| FiQA2018 | **0.4546** | 0.4394 | 0.4468 | 0.4108 | 0.4378 | 0.3328 | 0.3328 | 0.4240 |
| HotpotQA | 0.6814 | **0.7240** | 0.6304 | 0.6944 | 0.7121 | 0.6856 | 0.6508 | 0.7221 |
| MSMARCO | **0.4485** | 0.4404 | 0.3933 | 0.3831 | 0.4370 | 0.4229 | 0.4097 | 0.4203 |
| NFCorpus | 0.3533 | 0.3588 | 0.3638 | 0.3167 | 0.3406 | 0.3246 | 0.3100 | **0.3622** |
| NQ | 0.6365 | **0.6404** | 0.5990 | 0.5894 | 0.6302 | 0.5994 | 0.5623 | 0.6246 |
| QuoraRetrieval | **0.8901** | 0.8873 | 0.8802 | 0.8857 | 0.8817 | 0.8763 | 0.8694 | 0.8659 |
| SCIDOCS | 0.2024 | 0.2034 | 0.1821 | 0.1642 | 0.1746 | 0.1719 | 0.1346 | **0.2149** |
| SciFact | 0.7058 | 0.7180 | **0.7300** | 0.6419 | 0.7042 | 0.6950 | 0.6770 | 0.7159 |
| Touche2020 | 0.2950 | 0.2977 | 0.2346 | 0.2213 | 0.2314 | 0.2135 | 0.2117 | **0.3143** |
| TREC-COVID | 0.8387 | 0.8028 | 0.5722 | 0.5568 | 0.7123 | 0.6971 | 0.7119 | **0.8462** |

Table 4: Retrieval performance on MTEB Retrieval task (English). AE stands for Arctic-Embed.

## E    Replication of "Language Gap" with Fewer Pretraining Samples

We perform a comparison of several variations of our training recipe to confirm our intuition that we are not observing any significant "language gap" in our training. In a shortened training procedure with only 13K contrastive pretraining steps, we vary the following factors:

- MLM base model variants: English-only (En-GTE) vs. multilingual (mGTE);
- Pretraining data: English-only (English portion of our pretraining data only) vs. multilingual (original data mix);
- Fine-tuning data: English-only (English portion of our fine-tuning data) vs. multilingual (English fine-tuning data plus non-English MIRACL training set).

As shown by the results tabulated in Table 5, none of the multilingual treatments we attempted induce a sizable decrease in MTEB Retrieval score compared to the all-English

| MLM | PT | FT | Evaluation Sets | | | |
|-----|-----|-----|--------|------|--------|----------|
| | | | MTEB-R | CLEF | MIRACL | MIRACL-O |
| En | En | En | 0.526 | 0.327 | 0.114 | 0.271 |
| | | Mul | **0.532** | 0.340 | 0.268 | 0.361 |
| Mul | | En | 0.524 | 0.439 | 0.486 | 0.478 |
| | | Mul | **0.532** | 0.442 | 0.588 | 0.517 |
| | Mul | En | 0.525 | 0.451 | 0.530 | 0.521 |
| | | Mul | 0.529 | **0.452** | **0.594** | **0.538** |

Table 5: Impact of masked language modeling (MLM) base model, pretraining (PT), and finetuning (FT) data configurations with either English-only (En) or multilingual (Mul) content, showing their impact on downstream retrieval evaluations.

baseline. In fact, we actually observe a slight positive effect from including the non-English MIRACL finetuning datasets even on the English MLM base model.

