# OpenReview forum: "Arctic-Embed 2.0: Multilingual Retrieval Without Compromise"
_colmweb.org/COLM/2025/Conference — COLM 2025_

### Official Review · Reviewer_P4sz · 2025-05-13

**Rating:** 7
**Confidence:** 4
**Ethics Flag:** 1

**Summary:**

This paper introduces Open-Multilingual-Embed, compact multilingual dense retrievers trained using a three-stage pipeline (MLM, massive multilingual contrastive pretraining, and high-quality fine-tuning). The models integrate Matryoshka Representation Learning (MRL) for embedding compression, allowing a 3–4× smaller footprint with minimal performance loss. They achieve state-of-the-art open-source multilingual and English retrieval performance.

**Questions To Authors:**

- How does your method generalize to (non-European) non-latin or low-resource languages?
- Have you tested compression beyond 256 dimensions? Is 256 dims optimal?
- Do you plan to release code/scripts for reproducibility?

**Reasons To Accept:**

- Effective compression with MRL, significantly reducing embedding dimensions (768/1024→256).
- Strong empirical results across multilingual (CLEF) and English (MTEB-R) benchmarks.
- Training pipeline with massive multilingual data (1.41 B pairs).

**Reasons To Reject:**

- Limited language diversity (pretraining covers seven European languages).
- Limited ablations on embedding dimension choices or stage-specific MRL impact.
- High computational requirements (32 × H100 GPUs) limit reproducibility.

---

> ### Author Response · Authors · 2025-05-31
>
> Thank you for your detailed feedback! Please allow us to address your questions and comments.
>
> **Questions**:
>
> 1. How does your method generalize to (non-European) non-latin or low-resource languages?
>
> A substantial portion of the MIRACL benchmark includes non-Latin and low-resource languages, and our model performs reasonably well on these. That said, one of the key arguments in our paper is that MIRACL (based on Wikipedia) may be overfit by many existing models. We believe that performance on datasets outside the Wikipedia domain provides a more meaningful test of generalization. For example, we recently evaluated our model on the Arabic RAG dataset and observed strong performance, despite no task-specific training.
>
> 2. Have you tested compression beyond 256 dimensions? Is 256 dims optimal?
>
> We have previously done some informal testing of smaller dimensionality in text embedding models. Since dropping to 128 dimensions seemed to deliver a substantially bigger drop in retrieval scores than dropping to 256 in those experiments, we stuck with the safer option when designing Open-Multilingual-Embed.
>
> 3. Do you plan to release code/scripts for reproducibility?
>
> Yes. We have recently released code snippets for data processing and training; however, we are unable to include links here due to anonymization requirements. We will revise the final version of the paper to include these resources for full reproducibility.
>
>
> **Comments**:
>
> High computational requirements – While we fully embrace COLMs view that researchers should not be expected to have access to large-scale compute, we feel that the 32-GPU scale of our work should not be a reason to reject.

---

> > ### Comment · Reviewer_P4sz · 2025-06-10
> > **Response to Authors**
> >
> > Thank you for your response. I will keep my original score.

---

### Official Review · Reviewer_aVcC · 2025-05-13

**Rating:** 7
**Confidence:** 4
**Ethics Flag:** 1

**Summary:**

- The paper introduces Open-Multilingual-Embed models which report strong English (0.554 nDCG@10 on MTEB-R) and multilingual (0.534 nDCG@10 on CLEF) performance with a 113M parameter model
- It also reports that these models support Matryoshka Representation Learning (MRL), which allows for up to 4x compression with small quality degradation (retains about 99% of MTEB-R scores after truncation to 256 dimensions from the original 768)
- The models are trained on a dataset of 1.41B samples in 8 European languages (English, French, Spanish, German, Italian, Portuguese, and Polish) which the authors filter from various sources. They experiment with two models used for Hard Negative Mining and find that `gte-large-en-v1.5` works better than `stella-en-1.5B-v5` for this purpose
- Further experiments provide additional insights, such as
  1. refuting the view that pretraining on distant languages like Chinese harms English retrieval, since adding Chinese data actually improved MTEB-R English scores
  2. that pretrained checkpoint evaluations are shown to poorly predict final performance, because prolonged pretraining initially degraded but ultimately improved scores after finetuning
  3. that contrastive pretraining offers limited cross-lingual transfer to unrepresented languages, but fine-tuning significantly improves their performance after this pretraining phase
- A big portion of the paper is devoted to the "English performance gap", which the authors suggest may ultimately boil down to the lack of high quality multilingual training data.

**Reasons To Accept:**

- A solid practical contribution with a lot of potential impact
- A blueprint for training similar multilingual models
- Interesting experiments

**Reasons To Reject:**

1. The evaluation primarily relies on MTEB-R and CLEF and lacks ablation studies for various design choices such as the effect of different data filtering strategies and their parameters
2. Cross-lingual transfer is only measured for a narrow set of target languages (often high-resource), and the model's performance on an actual low-resource, unseen, or code-mixed languages is not investigated further
3. The experiments lack statistical significance tests, and it is hence difficult to assess to what extent can the gains be attributed to chance
4. Some of the important claims (e.g. adding adding Chinese data improved MTEB-R English scores) are not explored further and so they may be relatively easily attributed to dataset artifacts rather than to a new phenomena
5. The paper does not provide a rigorous error analysis or qualitative examples to demonstrate where the model fails or underperforms, which weakens the understanding of its limitations

---

> ### Author Response · Authors · 2025-05-31
>
> Thank you for your detailed feedback! Please allow us to address your questions and comments:
>
> **Questions**:
>
> 1. The evaluation primarily relies on MTEB-R and CLEF and lacks ablation studies for various design choices such as the effect of different data filtering strategies and their parameters
>
> We conducted substantial experimentation with different data filtering strategies during the early stages of model development. Unfortunately, however, we found that optimal parameters often varied significantly across languages and pre-training datasets, and the cost of repeated pretraining prohibited the thorough exploration necessary to present something scientifically sound on this topic.  Given this outcome, we did not feel it was feasible to center much of the paper on early-stage ablation studies, but we do agree that such research would be quite valuable to the scientific community.
>
> 2. Cross-lingual transfer is only measured for a narrow set of target languages (often high-resource), and the model's performance on an actual low-resource, unseen, or code-mixed languages is not investigated further
>
> We believe there may be some misunderstanding here. Our evaluation of cross-lingual transfer is conducted on the full MIRACL benchmark (Figure 4), which includes several low-resource languages such as Swahili, Yoruba, and Finnish. These languages are also potentially unseen in contrastive pre-training, as our data does not intentionally include them. The dotted lines in the figure represent zero-shot (non-finetuned) runs. Regarding code-mixed languages, we are not aware of existing retrieval benchmarks in this space, but we are open to exploring them if such datasets become available.
>
> 3. The experiments lack statistical significance tests, and it is hence difficult to assess to what extent can the gains be attributed to chance
>
> Thank you for flagging this. We will revise the paper to include statistical significance testing to strengthen the validity of our findings.
>
> 4. Some of the important claims (e.g. adding adding Chinese data improved MTEB-R English scores) are not explored further and so they may be relatively easily attributed to dataset artifacts rather than to a new phenomena
>
> We did not intend to claim that “adding any Chinese data improves MTEB-R English scores,” and we will edit the manuscript to more clearly present our results as a negative result disagreeing with the hypothesis that “adding Chinese data harms English performance”, which we view as a substantially weaker (and more reasonable) claim. Are there any other specific tenuous claims you have in mind which we can adjust to improve the paper in your eyes?
>
> 5. The paper does not provide a rigorous error analysis or qualitative examples to demonstrate where the model fails or underperforms, which weakens the understanding of its limitations
>
> We appreciate the suggestion! Do you have any suggestions for qualitative examples that would be helpful to you as a reader? There is always a fine line to walk between getting lost in the individual examples of an evaluation and finding true insight, and we would be happy to expand our appendix during revisions if there is something that would make the paper better.

---

> > ### Comment · Reviewer_aVcC · 2025-06-08
> >
> > > We did not intend to claim that “adding any Chinese data improves MTEB-R English scores,” and we will edit the manuscript to more clearly present our results as a negative result disagreeing with the hypothesis that “adding Chinese data harms English performance”, which we view as a substantially weaker (and more reasonable) claim. Are there any other specific tenuous claims you have in mind which we can adjust to improve the paper in your eyes?
> >
> > Thank you for your response. It might have been simply me misinterpreting the written text, for which I apologize. I don't think I ran across any other tenuous claims -- this was the only one that sort of stood out.
> >
> >
> > > Do you have any suggestions for qualitative examples that would be helpful to you as a reader? There is always a fine line to walk between getting lost in the individual examples of an evaluation and finding true insight, and we would be happy to expand our appendix during revisions if there is something that would make the paper better.
> >
> > I certainly agree that the line between a single individual example and actual insight is very thin and hence would only expect reasonable things to be done here.
> >
> > One idea that immediately comes to mind is investigating what kind of queries does the 1-2% of performance that is lost when truncating embedding represent. In other words, kinds of queries become more difficult after truncation?
> >
> > Perhaps these won't follow any discernible pattern but if they did and for instance it turned out that this drop represents queries that require understanding complex relationships or more abstract concepts, which might be encoded in the less significant dimensions of the embedding, this would significantly increase the paper's added value.

---

### Official Review · Reviewer_XDLM · 2025-05-20

**Rating:** 6
**Confidence:** 4
**Ethics Flag:** 1

**Summary:**

The authors release Open-Multilingual-Embed (OME), two open-source dense‐retrieval encoders (113M and 303M params) trained with a three-stage recipe—MLM, large-scale contrastive pre-training, and hard-negative fine-tuning using high-quality European-language data plus retrieval-based filtering—and compressed with Matryoshka Representation Learning (MRL). On MTEB-Retrieval and CLEF, OME either matches or surpasses open baselines and remains competitive with closed-source APIs.

The paper also reports exploratory studies on cross-lingual transfer (CLT) and the English performance gap. They found that evaluation during pre-training cannot accurately indicate performance after fine-tuning. They also found that after the first ~10 k steps, CLT declines for most families not in the pre-training data. For English performance gap, the authors demonstrate that in contrast to other models, OME does not exhibit the English performance gap (performance on English dropping after training on multilingual data). They speculate it is due to higher data quality that they curated via filtering, hard negative mining, stronger teacher etc.

**Reasons To Accept:**

1. OME achieves competitive accuracy without sacrificing English language performance, surpassing mE5-Base and BGE-M3 on retrieval benchmarks while avoiding the usual ≥3-pt English drop.

2. The paper consists of thorough and complete ablations -- Hard-negative mining thresholds, curriculum vs. random ordering, and cross-lingual transfer curves etc. This provides guidance to those looking to build their own multilingual encoders.

3. The commitment to open-source facilitates reproducibility and further research.

4. Given the complexity and nuances of multi-stage pipeline involved in training an encoder, the authors did a really good job of clearly communicating complex ideas. The paper is very well-written and reads nicely, with each component well motivated and linked back to prior work.

**Reasons To Reject:**

The work is primarily a tidy integration of known tricks. Every core component—MLM warm-start, InfoNCE contrastive pre-training, NV-Retriever style hard negatives, MRL compression is known techniques already in use for training encoders. The exploratory research questions (CLT and English gap) are good initial studies with speculative explanations but lack deep rigorous scientific exploration.

Due to this, I believe the paper is better suited as an industry or system track paper, rather than a main conference paper

---

> ### Author Response · Authors · 2025-05-31
>
> Thank you for your detailed feedback! Please allow us to address your questions and comments:
>
> 1. I believe the paper is better suited as an industry or system track paper, rather than a main conference paper
>
> We appreciate your perspective on the positioning of the paper. Interestingly, this work was previously submitted to the industry track of a top-tier venue and received very positive reviews. However, it was ultimately rejected due to a shared view among reviewers that the paper was “more appropriate for the research track” since it addresses the development of language models rather than the deployment of language models to specific applications in industry.
>
> When we saw the breadth of scientific and applied topics listed in the COLM call for papers (https://colmweb.org/cfp.html) and the multi-dimensional values espoused in the review guidelines (https://colmweb.org/ReviewGuide.html), we felt that our work fit right in with COLM. We are eager to better understand which aspects of our work may be misaligned with which aspects of the COLM CFP and Review Guide – would you mind taking some time during this discussion period to let us know in particular where you see the misalignment?

---

> > ### Comment · Reviewer_XDLM · 2025-06-10
> >
> > Thank you for your response and for sharing the context of your prior submission experience. I do agree that COLM encourages a broad range of contributions, spanning both applied and foundational work.
> >
> > However, the core concern I have is not about scope alignment with COLM, but rather about the contribution type and the level of novelty and insight. As mentioned before, the techniques integrated in the paper—MLM warm-start, InfoNCE-style contrastive learning, hard negative mining and MRL compression—are all individually well-established in the literature. The work does not substantially extend or deepen our understanding of these techniques or their interactions.
> >
> > Due to this, I choose to keep my original score.

---

### Official Review · Reviewer_HHZh · 2025-05-24

**Rating:** 7
**Confidence:** 4
**Ethics Flag:** 1

**Summary:**

The paper presents a well-engineered, open-source multilingual embedding model and provides valuable insights into multilingual training dynamics. The authors conduct a thorough investigation of training strategies, data filtering methods, and cross-lingual transfer effects, and also offer practical guidance for future work. While the empirical analysis is extensive, the exposition could be clearer, and the discussion of their proposed research questions would benefit from greater depth and alignment with the presented results.

**Questions To Authors:**

1. Adding per-task results for individual MTEB retrieval tasks in the appendix would increase transparency and enable more fine-grained analysis.

2. Although not critical to the current work, it would be interesting to explore whether multilingual retrieval quality can be improved using small LLM-based retrievers (e.g., 1.5B parameter scale).

**Reasons To Accept:**

1. The Open-Multilingual-Embed-L model achieves strong multilingual retrieval results on the CLEF benchmark while retaining competitive performance on English retrieval tasks, such as those in MTEB-R. This balance addresses the issue in multilingual embedding models, which often degrade in English-specific scenarios.

2. The authors’ commitment to open-source development enhances transparency and supports reproducibility, which is critical for further research in this domain.

3. The training pipeline is clearly explained, including detailed data filtering techniques, robust contrastive learning, and the integration of Matryoshka Representation Learning.

**Reasons To Reject:**

1. In Table 3, the empirical analysis could be more compelling with a fuller comparison across all benchmarked models, including OpenAI’s Text Embedding 3, Google’s Text Embedding 4, and Voyage Multilingual 2 on MIRACL, MIRACL-O, and CLEF. This is particularly important given the paper’s emphasis on generalization and the noted overfitting tendencies of some models to MIRACL.

2. The paper lacks a cohesive narrative that connects its methodology and main results to the research questions posed later on. Clarifying how the investigations into the "English performance gap" and "cross-lingual transfer" influence overall performance and inform methodological improvements would deepen the paper's contribution.

---

> ### Author Response · Authors · 2025-05-31
>
> Thank you for your detailed feedback! Please allow us to address your questions and comments.
>
> **Questions**:
>
> 1. Adding per-task results for individual MTEB retrieval tasks in the appendix would increase transparency and enable more fine-grained analysis.
>
> Great suggestion—we already have those results and will revise the paper to include them in the appendix.
>
> 2. Although not critical to the current work, it would be interesting to explore whether multilingual retrieval quality can be improved using small LLM-based retrievers (e.g., 1.5B parameter scale).
>
> We’re actually in the process of exploring this and hope to share results in a separate work soon.
>
>
> **Comments**:
>
> Missing empirical evaluations – It was unfortunately challenging to get the budget to run evaluations on closed-source models (it’s sadly a different budget for us than our GPU compute), but we agree that these experiments would be valuable to the community. We were glad to find a limited set of these evaluations had already been run and published in the Google Gecko paper (https://arxiv.org/abs/2403.20327), and if you are aware of any other works which may have published more of these numbers, please let us know so that we can include them for a more thorough comparison in our results section.

---

> > ### Comment · Reviewer_HHZh · 2025-06-01
> > **Reply to authors**
> >
> > I have read the comments by the authors. I hope the extra experiments would be added in the next versions. I will keep my scores.

---

### Official Review · Reviewer_wM18 · 2025-05-25

**Rating:** 5
**Confidence:** 5
**Ethics Flag:** 1

**Summary:**

The purpose of this paper is to propose a multilingual biencode that is efficient and effective. While the paper neglects to identify it, this is a first-stage retrieval that operates as a bi-encoder. In particular a new single-vector per passage model is trained that consists of fewer than 1B parameters. They gather 1.4B pre-training samples from 4 sources - Arctic Embed, mWiki, CC News, and MC4.  Some of the query-document pairs are filtered when the documents are what are associated with the queries is ranked lower than 20. The model is trained with contrastive loss. Hard negative mining is included.Embeddings are compressed with Matryoshka Representation Learning and scalar quantization.

The approach is evaluated on MTEB-R for English retrieval performance and CLEF 2000-2003 for cross language performance. Relative to the other single vector per passage approaches, this new model is competitive with the English performance and outperforms on CLEF.

**Questions To Authors:**

Q1 What is the performance on CLEF in terms of MAP?

Q2 Do the results observed on CLEF hold up on NeuCLIR?

**Reasons To Accept:**

Single-vector per passage approaches have been weaker in CLIR. Determining an effective approach to train such as a model is an important contribution.

The approach used to train the model for retrieval is clearly described.

**Reasons To Reject:**

The paper neglects to compare against state-of-the-art baselines for CLIR such as ColBERT with PLAID-X and Learned Sparse Representations with SPLADE. Both of these techniques are quite effective for both CLIR and monolingual retrieval and cannot be ignored. It is also surprising that NeuCLIR was not chosen as a benchmark over CLEF since it is much newer and better judged. Finally, nDCG is not the standard metric used on CLEF, which makes it challenging to compare the results reported to other prior work. MAP is the standard metric used for CLEF. It is also unclear why MSMARCO was not used as training data. Many CLIR approaches rely on mMarco and neuMARCO for such training.

The paper attempts to draw conclusions on English performance ability for cross-language by comparing scores; however, such a comparison is not particularly meaningful because even among the same collection documents different sets of queries will produce different scores.

---

> ### Author Response · Authors · 2025-05-31
>
> Thank you for your detailed feedback! Please allow us to address your questions and comments.
>
> **Questions**:
>
> 1. What is the performance on CLEF in terms of MAP?
>
> We did calculate MAP when conducting evaluations, however MAP highly correlated with nDCG and didn’t provide any additional insights worth including in the manuscript.
>
> 2. Do the results observed on CLEF hold up on NeuCLIR?
>
> We did not evaluate on NeuCLIR for two reasons. First, our work does not target cross-lingual information retrieval (i.e., queries in one language and documents in another); Second, our work primarily targets certain European languages – improving retrieval quality on languages like Chinese, Russian, and Persian is beyond the scope of this work.
>
> **Comments**:
>
> 1. Cross-lingual transfer (CLT) versus cross-lingual information retrieval (CLIR) – We would like to point out that Open-Multilingual-Embed primarily addresses CLT, not CLIR. While we appreciate your sharing your obvious expertise in CLIR with us, we are curious if you have any questions which might be more topically aligned with the focus of our work. If so, please share them during the discussion period!
>
> 2. Retrieval methods beyond dense single-vector retrieval – While sparse and multi-vector methods like ColBERT and SPLADE are certainly effective in numerous IR settings, their substantially different computational and storage profiles make it difficult to cleanly compare to dense single-vector methods like Open-Multilingual-Embed. Ultimately we felt that the precedent of other publications in the field of text embedding models (such as mE5 and GTE) was a reasonable one to follow, and we thus limited the focus of our evaluations to dense single-vector methods, but if you have examples of how embedding model papers have handled this balance better, please let us know during the discussion.

---

> > ### Comment · Reviewer_wM18 · 2025-06-07
> >
> > With respect to MAP v. nDCG, I agree they are highly correlated. The importance of using MAP is the fact the when others have used this data set, they have used MAP. By also reporting MAP, it makes this work comparable to other work that looked at monolingual retrieval across multiple languages.
> >
> > As for the problem you address, it would be helpful if you called out in the abstract and the introduction that you are interested in monolingual retrieval in multiple languages. The first paragraph of your introduction says "these multilingual text embedding models enable non-English monolingual retrieval as well as cross-lingual retrieval." I did not find a statement in the paper that said the queries were in the same language as the documents. Even multilingual retrieval is an ambious term as it does say which part of the problem is multilingual. For instance the documents could be in many languages and queries are in a single language.
> >
> > Given that it appears you are indexing each of the languages of CLEF separately and performing experiments in each language, would you report performance on each language separately, at least in the appendix to allow for by-language inspection of results.
> >
> > While I agree the NeuCLIR does not focus on the monolingual retrieval problem, queries are provided in each of the languages so that the dataset could be used to study the problem you are interested in.
> >
> > Finally, why did you decide not to use mMARCO for training which does support the type of retrieval you are experimenting with?

---

### Decision · Program_Chairs · 2025-07-08

**Decision:**

Accept

**Comment:**

The paper presents Open Multilingual Embed (OME), a compact multilingual bi-encoder for first-stage retrieval with under 1B parameters. Trained on 1.4B filtered query-document pairs using a three-stage pipeline and compressed with Matryoshka Representation Learning, OME achieves strong performance on both English (MTEB-R) and cross-lingual (CLEF) benchmarks. It outperforms comparable open models and avoids the typical English performance drop in multilingual training. The paper also provides insights into training strategies and cross-lingual transfer, though the exposition could be clearer and more tightly connected to its research questions.

The paper received positive feedback from the reviewers, who highlighted that the Open Multilingual Embed L model demonstrates strong performance on multilingual retrieval benchmarks such as CLEF, while also maintaining competitive results on English tasks in MTEB-R. This is particularly noteworthy given that multilingual models often experience a drop in English performance. The reviewers also commended the authors' commitment to open-source development, which enhances transparency and facilitates reproducibility—both crucial for progress in this field. Furthermore, the training pipeline is well-documented, with clear descriptions of data filtering strategies, contrastive learning, and the application of Matryoshka Representation Learning for compact and efficient embeddings. Given the paper’s solid contributions and well-executed methodology, I recommend it for acceptance.

Suggestions for improvements:
- As mentioned by one reviewer, please revise the paper to include statistical significance testing to strengthen the validity of our findings.